# India may need an additional metric to assess the endemicity of malaria in low surveillance districts

**Chander Prakash Yadav[1,2‡], Sanjeev Gupta[1‡], Praveen K. Bharti[1], Manju Rahi[2,3], Nafis Faizi** [1,4]**, Amit Sharma** [1,2,5]*

**1** ICMR-National Institute of Malaria Research (NIMR), New Delhi, India, **2** Academy of Scientific and Innovative Research (AcSIR), Ghaziabad, India, **3** Indian Council of Medical Research (ICMR), New Delhi, India, **4** Jawaharlal Nehru Medical College, AMU, Aligarh, India, **5** Molecular Medicine, International Centre for Genetic Engineering and Biotechnology (ICGEB), New Delhi, India

‡ CPY and SG share first authorship equally to this work.
* directornimr@gmail.com

**Data Availability Statement:** Malaria data for year 2017 and 2018 are in public domain that can be accessed here: District wise malaria data for 2017

## Abstract

India's National Framework for malaria elimination is essentially dependent on Annual Parasite Incidence (API). API is the primary criterion for classifying states and districts into different categories: intensified control, pre-elimination, elimination, prevention and re-establishment of malaria. However, API's validity is critically dependent on multiple factors, one such important factor is Annual Blood Examination Rate (ABER) and is often considered as indicator of operational efficiency. Therefore, the present study aimed to determine whether the API is a sufficiently good malaria index to assess malaria endemicity in India. An in-depth analysis of malaria data (2017–19) was done to determine the appropriateness of API as a sole indicator of malaria endemicity. We stratified the Indian districts into three strata based on Annual Blood Examination Rate (ABER): <5, 5.0–5.0, >15, further APIs was compared with Slide Positivity Rates (SPRs) using sign rank test, independently in each stratum. API and SPR were found comparable (p-value 0.323) in stratum 2 only. However, in the case of lower ABER (<5%, strata 1), the API was significantly lower than the SPR, and higher ABER (>15%), the API was found substantially higher than the SPR. Thus, ABER tunes the validity of API and should avoid to use as a single indicator of malaria endemicity. API is an appropriate measure of malaria endemicity in high and moderate transmission areas where surveillance is good (ABER≥5%). However, it is vitally dependent upon surveillance rate and other factors such as population size, the selection of individuals for malaria testing. Therefore, where surveillance is poor (<5%), we propose that API should be complemented with SPR and the number of cases. It will significantly aid the design and deployment of interventions in India.

(https://nvbdcp.gov.in/Doc/Annual-Report-2017.pdf) and District wise malaria data for 2018 (https://nvbdcp.gov.in/Doc/Annual-Report-2018.pdf). Malaria data for 2019 is not available online at present, but can be requested from National Center for Vector Borne Diseases Control (NCVBDC): National Center for Vector Borne Diseases Control (NCVBDC) 22, Sham Nath Marg, Delhi - 110054 (Landmark: Opp. I.P. College, Near Civil Line Metro Station) PHONE # 91–011–23967745, 23967780 Website: www.nvbdcp.gov.in/ E-MAIL: nvbdcp-mohfw@nic.in.

**Funding:** The authors received no specific funding for this work.

**Competing interests:** The authors have declared that no competing interests exist.

## Introduction

The global burden of malaria has declined substantially since 2015, and the World Health Organization (WHO) set a goal for 35 countries, including India, to eliminate malaria by 2030 [1]. Globally, 241 million malaria cases were estimated in 2020, compared to 227 million cases in 2019, showing an increase in cases due to disruption of services due to the COVID-19 pandemic [1]. The WHO's South-East Asia Region (WHO-SEAR) estimated nearly 5 million malaria cases and 9000 deaths in 2020. The report showed that India accounted for 83% of malaria cases and 82% of malaria deaths in the WHO-SEARO region [1]. Malaria intensity indices are greatly dependent on surveillance rate, which may be further influenced by many factors and thus may be misleading malaria endemicity. WHO estimated ~4.15 million malaria cases in India in 2020, which is ~ 22.2 times the number of ~0.18 million reported to the national malaria programme, in 2020 [1–3]. There was also a significant disparity in malaria deaths between WHO estimates (~7400) and those reported by the National Center for Vector-Borne Diseases Control (NCVBDC) [1, 3]. A study conducted by Kumar et al (2020) on the surveillance-based estimation of malaria burden in India estimated malaria incidence for the year 2015–16 to be ~4 times higher than one million registered by the national program, but three times lower than the World Health Organization's estimate of 13 million [4]. Various other studies have highlighted the difference between the estimated and confirmed malaria burden in India [4–6]. The reasons for the disparity could be many, and a well-recognized reason is the non-inclusion of data from healthcare providers other than the national programme in government and non-government space. A need to integrate these data sources and a common platform has been highlighted earlier [7]. Also, as India moves along the path of malaria elimination, it is urgently needed that paper-based aggregated surveillance system is transformed into case-based digital surveillance [8, 9]

In 2015, the World Health Assembly adopted the WHO's Global Technical Strategy (GTS) for malaria 2016–2030, with the goal of a malaria-free world [10], and set the ambitious target of reducing the global malaria burden by 90% by 2030. Further in 2017, WHO released a Regional Action Plan for 2017–2030 to make the Southeast Asia Region malaria-free [11]. In India, The National Center for Vector Borne Diseases Control (NCVBDC) is the government of India's nodal national body for the prevention and elimination of malaria [2]. Malaria control efforts are carried out through an established healthcare infrastructure and monitoring system. The healthcare system is divided into three levels: primary, secondary, and tertiary. Active and passive surveillance data are combined at all levels, from sub-center to primary health centre to district to state. Based on the number of positive cases of malaria identified by active and/or passive monitoring using microscopy or rapid diagnostics, the annual parasite incidence (API) [12], an indicator of malaria endemicity is calculated as

$$API = \frac{\text{Total positive}}{Total\ population} * 1000$$

The WHO adopted API, to categorise malaria transmission intensity, enabling each country to stratify their geographical units according to local malaria transmission [13]. In India, the national framework for malaria elimination (NFME) 2016–2030, API is the primary criterion to classify the Indian states into four categories [14]. Further, the National Strategic Plan (NSP 2017–22) was formulated with a goal to provide a clear roadmap of strategies and activities at a granular level, i.e., the district, which is the unit of implementation for the national malaria control programme [15]. The NSP aligns with the NFME and describes achieving elimination in a phased manner and monitoring its progress over 2017–22.

According to district-level API data for 2019, 32 districts from 11 Indian states were in Category 3, while ten districts from eight states were in Category 2, and the rest of the districts were in Category 1 or 0 (Fig A in S1 Text). Accordingly, surveillance policies and strategies differ depending on the elimination phases depending on each district's API. The intensified control phase aims to implement targeted interventions and detect potential outbreaks. On the other hand, the purpose of the elimination/pre-elimination phase is to investigate and classify each case and foci of malaria. The API of the sub-center (below the Primary Health Centre level) is used to prioritise vector control measures. For sub-centers having API>1, vector control measures consist of universal coverage with insecticide-treated bed nets. If not covered, then two indoor residual spray and larval control rounds. For sub-centres with API < 1 or no cases, foci-based adult vector control interventions are recommended [15].

Annual Blood Examination Rate (ABER) is the number of people receiving a parasitological test for malaria per unit population per year and serve as index of operational efficacy of the programme [16, 17]. The recommended ABER is 10% for monitoring malaria endemicity using API as an indicator [16, 17]. However, if the ABER is below or above the recommended level, population-referent measures like API should be viewed with great caution [18]. The slide positivity rate (SPR), defined as the number of laboratory-confirmed malaria cases per 100 suspected cases examined, is an alternative indicator for estimating temporal changes in malaria incidence [19]. Declining SPRs have been cited as evidence for successful malaria control interventions in Africa [19]. SPR was discovered to be a powerful predictor of malaria transmission by Bi Y et al. (2012), and it may be utilised to enhance malaria elimination planning and implementation [20]. Improved monitoring will aid in planning, budgeting, assessment, tracking control operations and disease trends. This will allow the national malaria control programme to get maximum impact on resources invested. Reliance on API alone may not be thus sufficient as a malaria metric that can track malaria control's progress (or otherwise). Therefore, with the aim of malaria elimination in India, to compare and contract malaria endemicity indicators API, ABER and SPR in India.

## Methodology

### Context

Currently, in India NFME and NSP documents provide a road map for India's advocacy and phased elimination of malaria. In NFME, Indian states with zero indigenous malaria cases are in Category 0 (prevention of re-establishment phase); states reporting API < 1 are in Category 1 (elimination phase); states reporting API <1 but some districts report API ≥1 are in category 2 (pre-elimination phase) and states with API≥1 are in category 3 (intensified control phase). In NSP, Indian districts were stratified into four categories, i.e., intensified control phase (Category 3 if API ≥ 2), pre-elimination phase (Category 2 if 1≤API<2), elimination phase (Category 1 if API < 1), and prevention and re-establishment phase (Category 0, no local transmission and reporting no case for last three years) based on reported API. The goal of the National Strategic Plan was to eliminate malaria in category 1 and 2 districts by 2020 and 2022, respectively, and to reduce transmission in Category 3 districts to stable API<1 by 2022.

### Data sources

The Directorate of the National Center for Vector Borne Diseases Control (NCVBDC) is a central nodal agency for preventing and controlling six vector-borne diseases in India, and malaria is one of them [2]. The NCVBDC is responsible for collecting/recording the malaria data on various parameters from village to country level. NCVBDC's data on three

malariometric parameters: API, SPR, and ABER for the years 2017 to 2019 at the district level were analysed in this research article.

## Data analysis

District-wise data on three malaria parameters (i.e. API, SPR and ABER) were available for three consecutive years (2017–19). First, central values (median) for all malaria parameters: ABER, SPR, and API across three years (2017–19) were calculated. Then district API was independently compared with SPR in three strata of ABER independently (given below).

*Strata 1*: **ABER <5**
*Strata 2*: **5.0≤ABER ≤15.0**
*Strata 3*: **15<ABER**

These strata were constructed using the relationship of API, SPR, and ABER (Given in Eq 1). As per Eq 1, API is directly proportional to ABER, while SPR is inversely proportional to ABER as

$$API = \frac{SPR * ABER}{10} \tag{Eq 1}$$

As the ABER increases, SPR decreases and API increases. API perfectly correlates with SPR when the ABER equals 10% (given below):

$$if\ ABER\ is\ 10\% \Rightarrow API = \frac{SPR * 10}{10} = SPR$$

Further, if we assume the actual prevalence of fever in the community to be ~10%, then ABER should be equal to 10% ± margin of error; if we again assume a margin of error as 5%, then ABER must be ranged in between 5% to 15% to get comparable API and SPR. Therefore, the three strata of ABER (i.e. <5% (strata 1), 5–15% (strata 2) and >15% (strata 3)) were constructed to test this hypothesis on actual data. The district API was compared with district SPR using the Sign-rank test independently in each stratum. However, the authors are aware that using the Sign-rank test in this situation may not be prudent since instead of comparing one variable in two related groups, we have compared two variables, i.e., API and SPR. It was assumed that both SPR and API measure the same thing, i.e., the intensity of malaria, which is influenced by the district's background characteristics.

All data analyses were carried out in statistical software R 3.4.4 and Stata 15.0. while geographical maps were constructed Using Quantum geographic information system (QGIS), an open-source software [21]. The shapefiles (.shp) of the base layers, namely state and district, were procured from survey of India, and all maps were produced at the host institute [22]. A p-value <0.05 was considered as statistically significant.

## Results

While analyzing the national programme (NCVBDC) data, we found that ABER distribution across districts was not uniform and varied significantly across India. A minimum of 10% ABER for all the districts was not attained for any of the years analyzed here. For example, in the year 2019, the average ABER of the country was 11.7% (SD = 8.0), and it varied from 0.01% to 58.8%. When, we mapped the ABER (median (2017–19)) for all districts of India; among 686 districts (for which ABER data was available), there were 152 (22.2%) districts where the ABER was lower than 5%, 376 districts (54.86%) where the ABER was reasonably good (varied from 5% to 15%), and in 158 districts (23.0%), ABER was high (i.e., ≥15%). In nearly half of the districts {(152+158) i.e. 45.2%}, the ABER is either much lower or higher

**Table 1. Comparison of district API and SPR in three strata of ABER.**

| Stata | N | API | | | SPR | | | P-value |
|---|---|---|---|---|---|---|---|---|
| | | $\bar{X} \pm \sigma$ | Median (P25 to P75) | Min to Max | $\bar{X} \pm \sigma$ | Median (p25 to p75) | Min to Max | |
| ABER<5 | 152 | 0.18±0.77 | 0.01 (0.004 to 0.096) | 0 to 7.31 | 0.80±1.8 | 0.23 (0.05 to 0.82) | 0 to 14.9 | <0.001 |
| 5≤ABER≤15 | 376 | 0.21±0.53 | 0.04 (0.01 to 0.16) | 0 to 3.87 | 0.22±0.56 | 0.04 (0.01 to 0.16) | 0 to 4.24 | 0.323 |
| ABER>15 | 158 | 2.43±7.46 | 0.15 (0.04 to 0.77) | 0 to 53.1 | 0.93±2.37 | 0.07 (0.02 to 0.46) | 0 to 14.65 | <0.001 |

API- Median Annual Parasite Incidence (2017–19); SPR- Median Slide Positivity Rate (2017–19)

$\bar{X}$ − Airthmetic mean; σ-Standard deviation; P25- 25[th] percentile; P75-75[th] percentile; Min-Minimum, Max-Maximum

than the recommended ones. This number is quite big and important as we know ABER directly affects API (Fig B in S1 Text). Therefore, to study this phenomenon in more detail, we compared district API (median (2017–19)) with district SPR (median (2017–19)) statistically in the above discussed three strata of ABER (<5, 5–15 and >15).

In the first strata of ABER (i.e. ABER≤5%), API was found significantly lower than the SPR when compared on absolute number (p-value <0.001, Table 1 and Fig 1). When the same was investigated in the context of National Strategic Plan (NSP)'s categorization, out of 152 districts in strata 1, there were only five districts where API was one or more (3 districts had API in between 1 to 2 and 2 districts had more than 2 API), however, there were 35 districts where SPR was one or more (22 districts had SPR between 1 to 2, and 13 had more than 2 SPR). It implies that if we go by API only, only five districts will get special attention in terms of malaria intervention, while if we add SPR along API, this number may go up to 35 districts (Table 2 and Fig 2).

In the second strata of ABER (5≤ABER≤15), API and SPR were found comparable when compared on absolute number (p-value 0.323, Table 1 and Fig 1). Similarly, no difference in the number of districts was observed when compared as per National Strategic Plan's categorization. By API alone, we identified 20 districts with API>1 (9 districts had API in between 1 to 2, and 11 had more than 2 API). In comparison, the addition of SPR with API does not add much to this number (21 districts: 9 districts had SPR in between 1 to 2 and 13 had more than 2 SPR) (Table 2 and Fig 3).

In the last strata (ABER>15%), the API was found to be significantly higher than SPR on absolute number (p-value <0.001, Table 1, Fig 1), and there was a significant difference in API (9 districts with API 1 to 2 and 27 with more than 2 API) and SPR (9 districts with SPR 1 to 2 and 19 with more than 2) as per NSP' categorization as well. In this strata, API itself can identify more malaria-endemic districts, and the addition of SPR does not add any new districts further (Table 2 and Fig 4).

In summary, the analysis suggests that except for districts belonging to the second strata-2 (where ABER is between 5 to 15), API and SPR were incomparable in all districts belonging to the strata 1 & strata 3. Districts where ABER was lower than 5%, API was found to be significantly lower than the SPR, and on the other hand, where ABER was more than 15%, the situation gets reversed, and we got a substantially higher API than SPR.

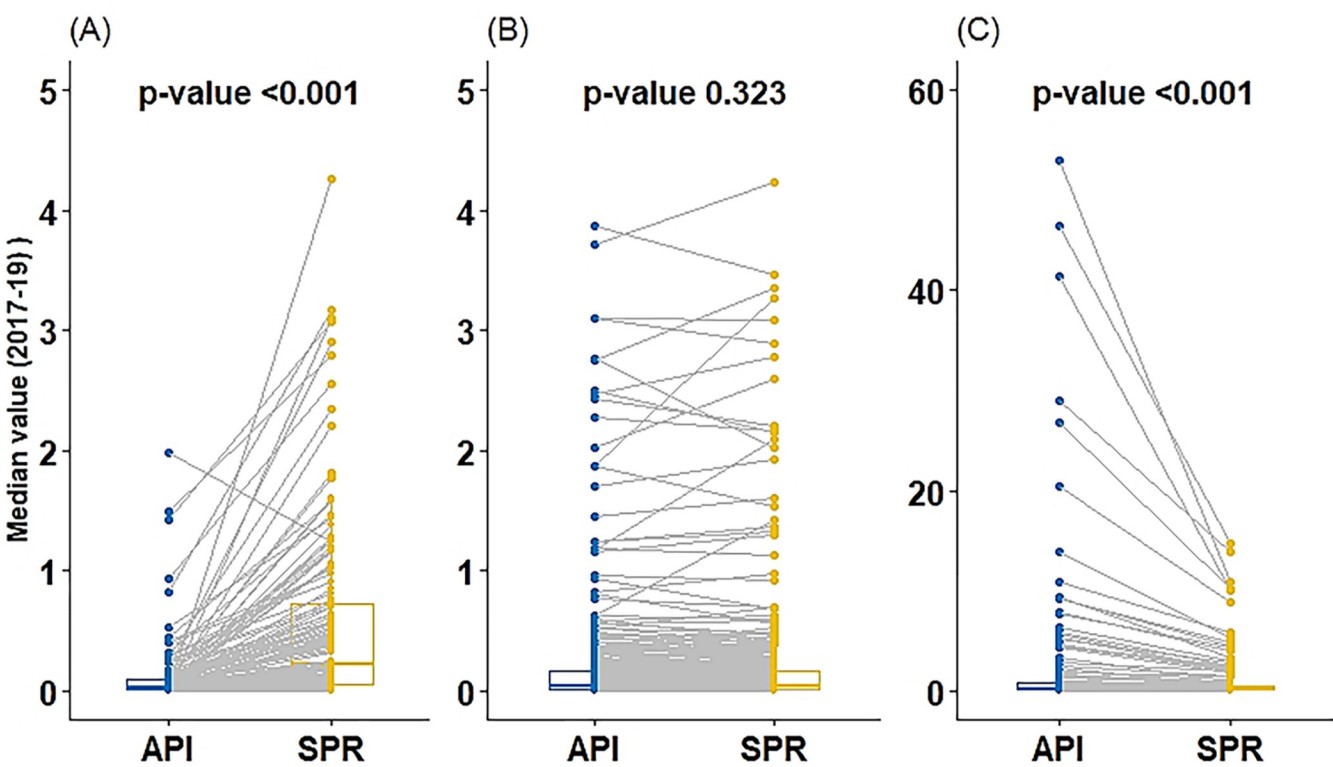

**Fig 1. Comparison of API and SPR in three stratums of ABER based on district-level data of India (686 districts).** API: Median Annual Parasite Incidence (2017–19); SPR: Median Slide Positivity Rate (2017–19) (A): Comparison of API and SPR in first stratum of ABER (i.e., ABER<5) (B): Comparison of API and SPR in second stratum of ABER (i.e., 5≤ABER≤15) (C): Comparison of API and SPR in third stratum of ABER (i.e., ABER>15).

The analysis also revealed that there are few states of India: Uttarakhand (Median (p25 to p75) ABER is 2.5 (1.5 to 3.66)), Uttar Pradesh (Median (p25 to p75) ABER is 2.3 (1.4 to 3.2)), Bihar (Median (p25 to p75) ABER is 0.15 (0.03 to 0.32)) and Sikkim (Median (p25 to p75) ABER is 2.14 (1.81 to 5.47)) along with these some regions of Himachal Pradesh and Jammu and Kashmir where last the three-years (2017–19) median ABER is below 5%. Amongst these,

**Table 2. Distribution of API and SPR as per National Strategic Plan (NSP 2017–22) in three strata of ABER.**

| Strata | API | | | SPR | | |
|---|---|---|---|---|---|---|
| | <1 | 1–2 | >2 | <1 | 1–2 | >2 |
| | elimination phase | pre-elimination phase | intensified control phase | elimination phase | pre-elimination phase | intensified control phase |
| **ABER<5 (N = 152)** | 147 | 3 | 2 | 117 | 22 | 13 |
| | (96.7%) | (2.0%) | (1.3%) | (76.9%) | (14.5%) | (8.5%) |
| **5≤ABER≤15 (N = 376)** | 356 | 9 | 11 | 354 | 9 | 13 |
| | (94.7%) | (2.4%) | (2.9%) | (94.1%) | (2.4%) | (3.5%) |
| **ABER>15 (N = 158)** | 122 | 9 | 27 | 130 | 9 | 19 |
| | (77.2%) | (5.7%) | (17.1%) | (82.3%) | (5.7%) | (12.0%) |

Row Percentages are given

API- Median Annual Parasite Incidence (2017–19) categorization as per NSP 2017–22

SPR- Median Slide Positivity Rate (2017–19) categorization as per NSP 2017–22

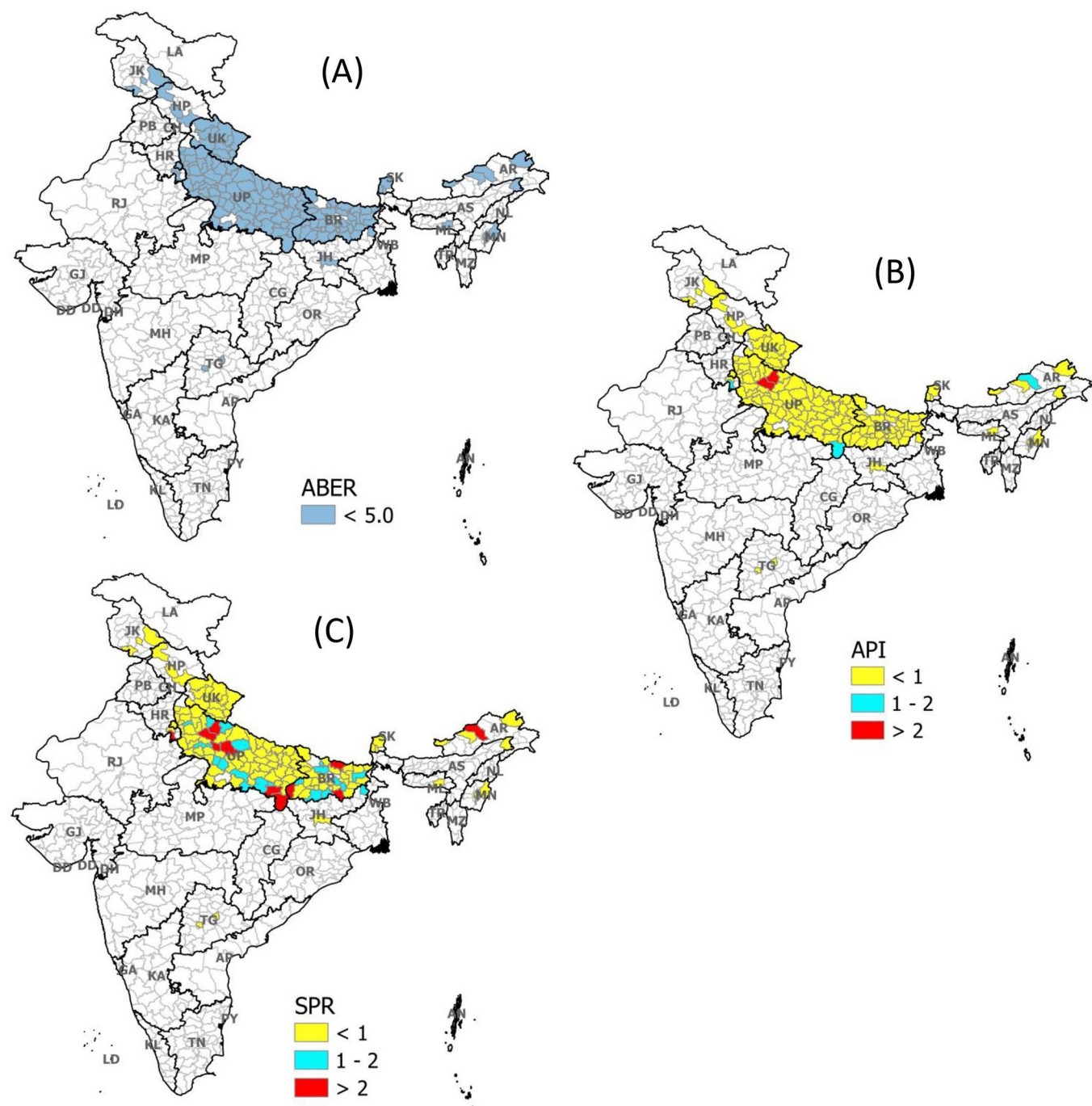

**Fig 2. Distribution of API and SPR among low surveillance districts of India.** (A): Representing all districts of India (blue shaded) where ABER <5 (Strata 1) (B): Distribution of API as per NSP in these districts (Strata 1) (C): Distribution of SPR as per NSP in these districts (Strata 1). ABER: Median Annual Blood Examination Rate across three years (2017–19); API: Median Annual Parasite Incidence across three years (2017–19); SPR: Median Slide Positivity Rate across three years (2017–19); NSP: National Strategic Plan 2017–22.

Bihar and Uttar Pradesh are important to study, in Bihar, more than 75% of districts (29 out of 38) having ABER lower than 0.32%, with a maximum ABER being only 0.54%. The result of this poor surveillance was that there was a huge difference (p<0.001) between API [Median (P25 to P75): 0.005 (0.001 to 0.012)] and SPR [Median (P25 to P75): 0.93 (0.19 to 1.25)],

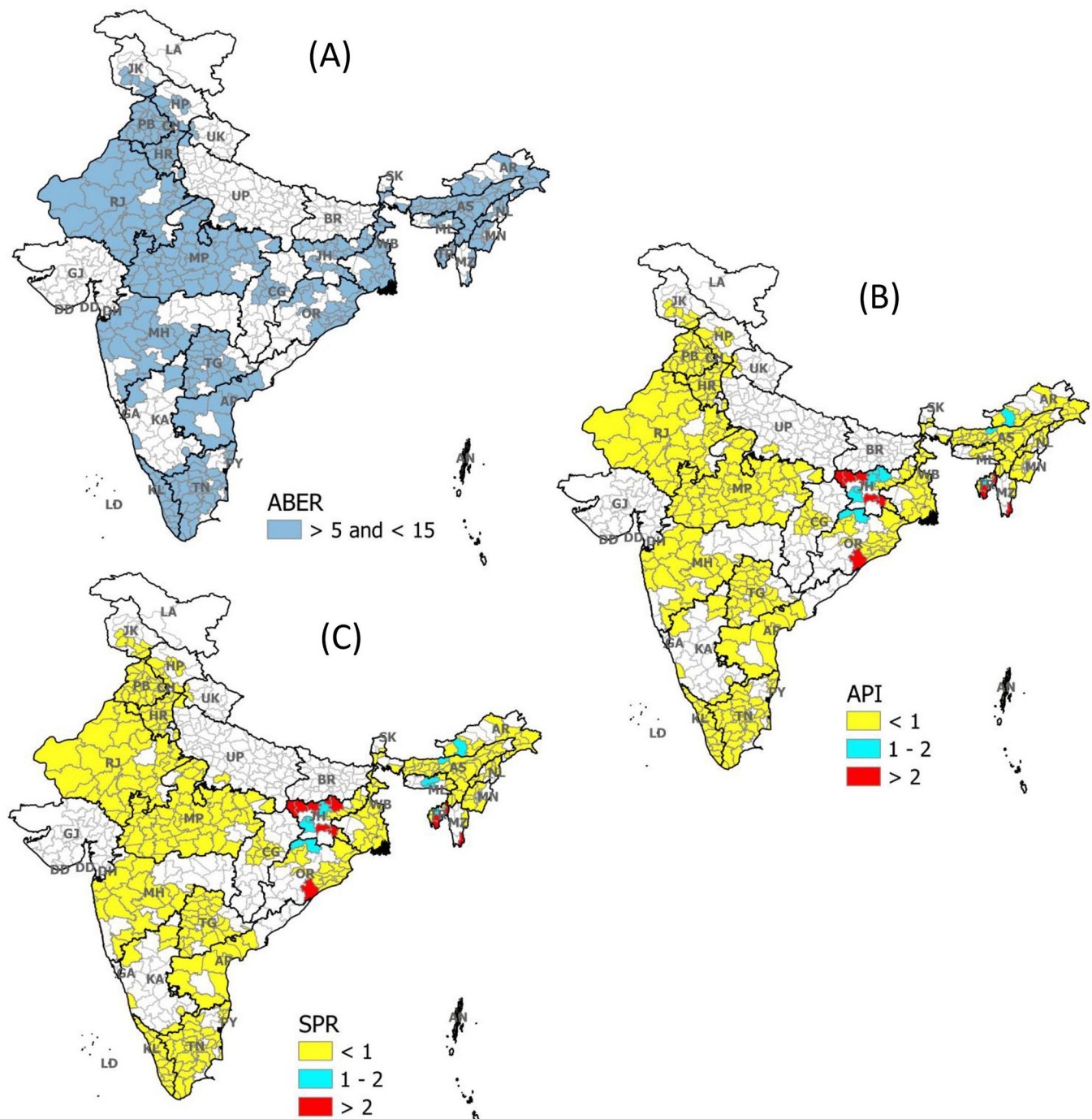

**Fig 3. Distribution of API and SPR among moderate surveillance districts of India.** (A): Representing all districts (blue shaded) of India where ABER ranges from 5 to 15 (Strata 2) (B): Distribution of API as per NSP in these districts (Strata 2) (C): Distribution of SPR as per NSP in these districts (Strata 2)). ABER: Median Annual Blood Examination Rate across three years (2017–19); API: Median Annual Parasite Incidence across three years (2017–19); SPR: Median Slide Positivity Rate across three years (2017–19); NSP: National Strategic Plan 2017–22.

probably API artificially decreased malaria endemicity due to lower surveillance rate. The situation in Uttar Pradesh was similar to Bihar in this context, where most districts (75%, 56 out of 75) fall under 3.24% ABER category, and the maximum ABER is only 6.46%. Here again, due

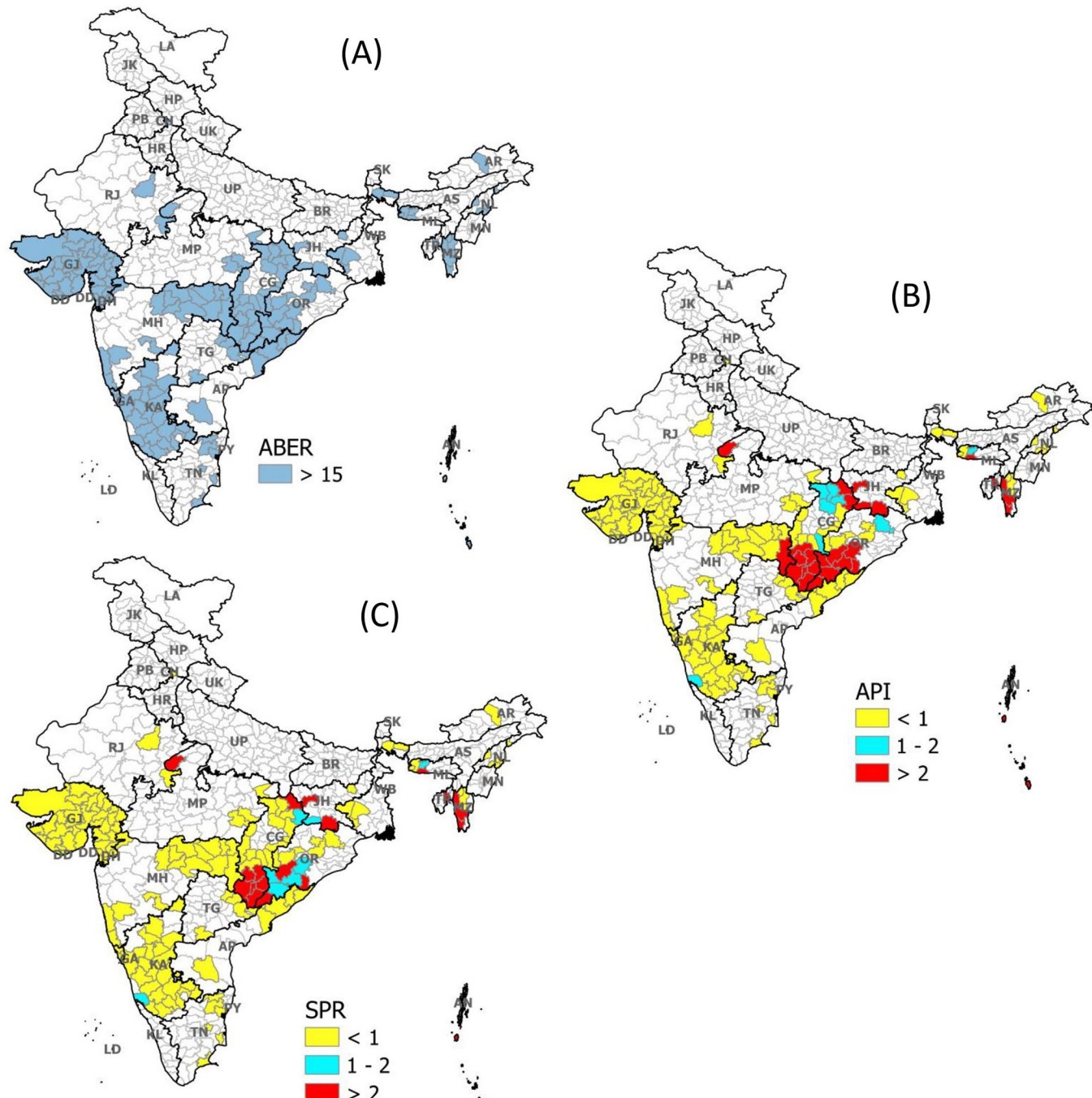

**Fig 4. Distribution of API and SPR among high surveillance districts of India.** (A): Representing all districts (blue shaded) of India where ABER >15 (Strata 3) (B): Distribution of API as per NSP in these districts (Strata 3) (C): Distribution of SPR as per NSP in these districts (Strata 3). ABER: Median Annual Blood Examination Rate across three years (2017–19); API: Median Annual Parasite Incidence across three years (2017–19); SPR: Median Slide Positivity Rate across three years (2017–19); NSP: National Strategic Plan 2017–22.

to lower ABER, API [Median (P25 to P75): 0.16 (0.07 to 0.16)] values were found significantly lower (p-value <0.001) than the SPR values [Median (P25 to P75): 0.32 (0.08 to 0.73)].

Similar to low surveillance (ABER<5) states, there were many states wherein most districts had a three-year average ABER of >15% such as Gujarat (Median (p25 to p75): 23.2 (20.6 to

24.9)), Karnataka (Median (p25 to p75): 18.8 (15.8 to 22.4)), Chhattisgarh (Median (p25 to p75): 19.8 (12.5 to 25.7)), Odisha (Median (p25 to p75): 14.4 (11.0 to 18.9)) and Maharashtra (Median (p25 to p75): 14.4 (11.6 to 18.0)). Gujarat, Karnataka and Chhattisgarh are crucial as their median ABER was more than 15%, which means that 50% of districts in these states have ABER 15 or more. In Gujarat, the minimum ABER (median (2017–19)) was ~18.9, because of such high ABER, relatively higher API (Median (p25 to p75): 0.14 (0.09 to 0.23)) as compared to SPR (Median (p25 to p75): 0.06 (0.05 to 0.06)) was observed. Though there was a difference in API and SPR due to higher ABER, there was still no change in intervention deployment because none of the districts had API or SPR of one or more. In a situation where endemicity is too low despite good surveillance, policymakers should focus more on the total number of malaria cases than on API and SPR. Like Gujarat, none of the districts except one in Karnataka had API or SPR one and above. The next state in the high surveillance category was Chhattisgarh which was highly malaria-endemic, and surveillance was excellent (average ABER is ~21.6); as a result, there was a significant difference between API (Median (p25 to p75): 1.58 (0.77 to 6.20)) and SPR (Median (p25 to p75): 0.82 (0.52 to 0.85)). If we see this data as per NSP categorization, there were 14 districts where API>1; and eight with SPR>1.

In order to provide more information on API, SPR and ABER relationship, the raw data of top 20 districts from each of the strata have been provided as supplementary material (Tables A–C in S1 Text: S1 Table describes the top 20 districts from the strata 1 (ABER<5) where the difference between API and SPR is maximum. Similarly, S2 and S3 Tables show data of top 20 districts from strata 2 and strata 3).

## Discussion

The present study aimed to determine whether the API is a sufficiently good malaria metric index to assess malaria endemicity as it highly depends upon the ABER. This analysis has focused on evaluating the role of SPR along with API for determining the malaria endemicity in different surveillance (or ABER) district. Assuming febrile illness in the community to be ~10%, and further assuming a margin of error as 5%, then ABER should vary between 5 to 15, as discussed in detail in the methodology. If this is the scenario on the ground, the estimate of malaria using API can be considered valid as there was no statistical difference observed between API and SPR. On the other hand, if we keep ABER below the true prevalence of fever minus the margin of error (i.e., ABER<5%), the malaria estimate by API may get artificially decreased than the actual estimate as there is no concordance between API and SPR (SPR> API). Similarly, API estimates may get artificially increased than the true value when ABER exceeds the true prevalence of fever plus the margin of error (i.e., ABER>15%). Thus, overestimation or underestimation increases as ABER deviates from the true prevalence of fever, plus or minus the margin of error (Fig 5). API alone is not enough to estimate malaria intensity because a small numerator (number of confirmed malaria cases) due to lower ABER is diluted in many situations by the high denominator (the population size). In such cases, API does not measure malaria transmission as accurately as it is supposed to measure [20]. Other than this, there may be error in estimating population size every year as census has taken place every 10[th] year only in India, which may further affect the validity of API. This limitation is easily observed when we compare the number of malaria cases in Phulbani and Ganjam (districts of Odisha) (Table 3). Here, we observe a considerable variation in API value due to varying population size and ABER rate, while malaria cases are more or less comparable in these two districts. While on the other hand SPR in these two districts is comparable to total cases and depicts a better picture. But stratification based on API puts Phulbani in the pre-elimination phase (Category 2 with API 1–2) and Ganjam in the elimination phase district (Category 1

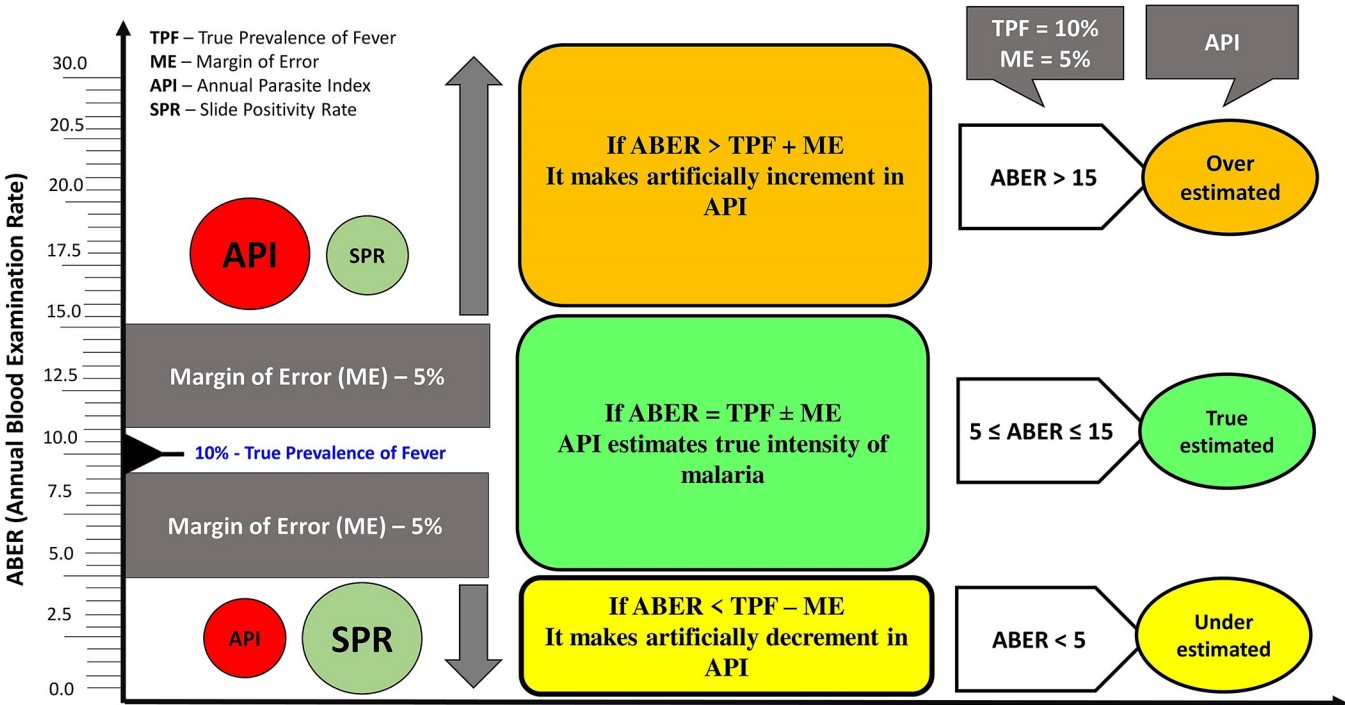

**Fig 5. Schema of relationship between API, SPR, ABER and true prevalence of fever.**

with API <1) despite Ganjam having a slightly higher number of cases. Even Koraput (district of Odisha) and Naraynpur (district of Chhattisgarh) have similar caseloads but considerable variation in API due to varying population size and ABER; on the other hand, SPR seems to be quite robust for these districts as a metric (Table 3). It is evident that the difference in population sizes and ABER can often lead to misleading APIs and thus derail the deployment of the right set of interventional tools. Tracking of malaria cases becomes important since the goal of malaria elimination is zero indigenous cases and not some minimum number of cases per 1000 population as for other diseases like visceral leishmaniasis (< 1 case per 10,000) [23].

Several studies have indicated that SPR can be a good surrogate marker of malaria transmission [19, 20, 24]. At the same time, we take into cognisance the limitations of SPR. A robust SPR is dependent on quality laboratory diagnostic testing, which may not be present, especially in microscopy, in all malaria-endemic settings in India. Moreover, when the criteria of surveillance (the denominator of ABER) is fever case, any change in the incidence of febrile illness

**Table 3. Comparison of API and SPR with respect to population size and ABER.**

| S.No | State | Year | District | Population in 1000 | Total cases | ABER | API | SPR | District Categorization |
|------|-------|------|----------|---------------------|--------------|------|------|------|--------------------------|
| 1 | Odisha | 2019 | Phulbani | 844 | 2978 | 31.1 | 3.5 | 1.1 | 3 |
|   |        |      |          |     |      |      |     |     | (Intensified Control Phase) |
| 1 | Odisha | 2019 | Ganjam | 3779 | 3390 | 10.9 | 0.9 | 0.8 | 1 |
|   |        |      |          |      |      |      |     |     | (Elimination Phase) |
| 2 | Odisha | 2019 | Koraput | 1492 | 4619 | 28.2 | 3.1 | 1.1 | 1 |
|   |        |      |          |      |      |      |     |     | (Elimination Phase) |
| 2 | Chhattisgarh | 2019 | Naraynpur | 160 | 4294 | 42.5 | 26.8 | 6.3 | 3 |
|   |        |      |          |     |      |      |      |     | (Intensified Control Phase) |

(fever due to any reason) will alter SPR in the absence of true change in the malaria incidence in the community. For example, fever is a common manifestation of SARS-CoV2 infection; hence applying malaria diagnosis to all these fever cases will artificially reduce SPR. Also, the age structure of the community and seasonality can influence the SPR either way. SPR is helpful in observing temporal trends in malaria incidence but cannot reflect actual incidence in population [19].

Despite the above limitations, SPR can be used along with API to fill the gap in defining malaria transmission, especially in those situations where ABER is too high (>35) or too low (<2.5). Interestingly, SPR has been the main indicator for malaria elimination in China [20], which has recently been declared malaria-free. SPR's use as an indicator has been consistent with malaria incidence decline in the island of Príncipe [25], Uganda [19] and recently China [20]. Thus, each Indian district should be defined by its SPR, API and caseload in order to convey a complete picture of its malaria transmission status.

## Conclusion

Countries implementing malaria control interventions should adopt a comprehensive model for classifying their regions (districts) to enable decision-making and deployment of interventional tools. Malaria surveillance remains a vital facet of any control or elimination goal. It relies heavily on a robust assessment of malaria incidence that any singular metric may not provide, and API certainly does not. API may be a good measure of malaria transmission in high and moderate transmission risk areas where ABER and fever surveillance is done very robustly. Malaria is a local and focal disease, so its cases are not distributed evenly in the population. Other than this, it is also not easy to correctly estimate population every year as census happens only every tenth year. For this reason, defining malaria transmission by taking the whole population as the denominator is not appropriate. Therefore, SPR and caseload must be used alongside API to facilitate malaria control, such as here: Endemicity of a district may be represented as $^{Cases}District\ Name.^{API}_{SPR}$. We suggest a re-stratification of Indian districts as a function of their SPRs and caseload alongside the usage of API as a new nomenclature to reassess the assignment of districts into various elimination categories.

## Supporting information

**S1 Text.**
(DOCX)

## Acknowledgments

We are very thankful to Directorate of NCVBDC for providing data and ICMR-NIMR for all logistical support.

## Author Contributions

**Conceptualization:** Chander Prakash Yadav, Manju Rahi, Amit Sharma.

**Data curation:** Chander Prakash Yadav, Sanjeev Gupta.

**Formal analysis:** Chander Prakash Yadav, Sanjeev Gupta, Praveen K. Bharti, Nafis Faizi, Amit Sharma.

**Investigation:** Chander Prakash Yadav, Praveen K. Bharti, Nafis Faizi, Amit Sharma.

**Methodology:** Chander Prakash Yadav, Praveen K. Bharti, Manju Rahi, Nafis Faizi, Amit Sharma.

**Project administration:** Chander Prakash Yadav.

**Software:** Chander Prakash Yadav.

**Supervision:** Amit Sharma.

**Validation:** Chander Prakash Yadav, Manju Rahi, Amit Sharma.

**Visualization:** Chander Prakash Yadav, Sanjeev Gupta.

**Writing – original draft:** Chander Prakash Yadav, Sanjeev Gupta.

**Writing – review & editing:** Praveen K. Bharti, Manju Rahi, Nafis Faizi, Amit Sharma.

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
