## [Decision Letter · Decision Letter 0]

4 Jul 2022

PGPH-D-22-00177

India may need an additional metric to assess the endemicity of Malaria and plan its elimination

Dear Dr. Sharma,

Thank you for submitting your manuscript to PLOS Global Public Health. After careful consideration, we feel that it has merit but does not fully meet PLOS Global Public Health’s publication criteria as it currently stands. Therefore, we invite you to submit a revised version of the manuscript that addresses the points raised during the review process.

We look forward to receiving your revised manuscript.

Kind regards,

Louisa Alexandra Messenger, MSc, PhD

Academic Editor

Journal Requirements:

1. Figures 2-4: please (a) provide a direct link to the base layer of the map used and ensure this is also included in the figure legend; (b) provide a link to the terms of use / license information for the base layer. We cannot publish proprietary or copyrighted maps (e.g. Google Maps, Mapquest) and the terms of use for your map base layer must be compatible with our CC-BY 4.0 license. 

2. In the online submission form, you indicated that "Data is available and may be shared on request.". All PLOS journals now require all data underlying the findings described in their manuscript to be freely available to other researchers, either 1. In a public repository, 2. Within the manuscript itself, or 3. Uploaded as supplementary information.

3. Please provide separate figure files in .tif or .eps format.

4. We have noticed that you have uploaded Supporting Information files, but you have not included a list of legends. Please add a full list of legends for your Supporting Information files after the references list. 

Additional Editor Comments (if provided):

Reviewers' comments:

Reviewer's Responses to Questions

**Comments to the Author**

1. Does this manuscript meet PLOS Global Public Health’s publication criteria? Is the manuscript technically sound, and do the data support the conclusions? The manuscript must describe methodologically and ethically rigorous research with conclusions that are appropriately drawn based on the data presented.

Reviewer #1: Yes

Reviewer #2: Partly

2. Has the statistical analysis been performed appropriately and rigorously?

Reviewer #1: Yes

Reviewer #2: No

3. Have the authors made all data underlying the findings in their manuscript fully available (please refer to the Data Availability Statement at the start of the manuscript PDF file)?

Reviewer #1: Yes

Reviewer #2: Yes

4. Is the manuscript presented in an intelligible fashion and written in standard English?

Reviewer #1: Yes

Reviewer #2: Yes

5. Review Comments to the Author

Reviewer #1: A well-written work presents the methods for incorporating SPR, ABER, and API parameters into categorizing malaria regions. I have no other comments on this paper.

Additionally, https://www.who.int/docs/default-source/searo/india/publications/national-framework-malaria-elimination-india-2016-2030.pdf?sfvrsn=c3e0ee29_4 , mainly Annex 7 Table on Malaria epidemiological situation for elimination planning by category used the author's formula: API = APR * ABER / 10.

Reviewer #2: The authors present an interesting concept of comparing different malaria indicators, that are used to inform targets and interventions. I have some concerns with their approach in terms of comparison and also several other minor comments:

Major comment

- It is not clear to me why the authors used the approach that they did for the comparison of the indicators (p 5&6). It would be much better to create a series of scatterplots to compare the correlation between the indicators and also summary measures of correlation. With that information, the authors could then be more informed in how the measures differed/were similar and also to develop thresholds if/when the indicators are comparable.

Minor comments

- Abstract

- Operational efficient indicator - would be more clear to remove and to first present the indicator and then explain how it's use is central to malaria elimination in India

- Finding - "...upon surveillance and other factors". Name a few of the other factors and it's really the quality of the surveillance program (vs surveillance)

- Finding - "...determines API and therefore latter is not..." This sentence is confusing and please be more direct (the API is not a robust indicator for malaria surveillance)

Title

- consider renaming the article to something that is more descriptive (e.g., comparing different malaria indicators for malaria elimination in India)

Introduction

- Malaria doesn't need to be capitalized (line 52)

- dependent on the quality of the surveillance program vs surveillance (line59)

- spell out abbreviations (line 63, line 79)

- I am confused as the API is presented as a rate (number of new confirmed malaria per 1,000) but then the thresholds are 0,1,etc. How is the API calculated then?

- the paragraph (lines 82-96) with the thresholds for API and categories is confusing. Would be helpful to explain the overlap between the API threshold and how it's decided to be in what phase (why certain states with the same API level in different elimination phases?)

- it would be better to start with a general introduction of the API and the elimination phases vs all of the details provided in the two paragraphs (lines 82-107)

- more information about how the 10% threshold of ABER was chosen (and this needs to be referenced) and the same for the API

- "Reliance on API alone...." (line 119-120) - this is what you're testing so I would word it differently as you don't know at this point if API alone is insufficient

- was vs is (line 121) - as the study has already been conducted. And try and reframe the objectives to be more precise (e.g., to evaluate the correlation between different malaria indicators (API, ABER, and SPR) for 2016-2019 in India)

- the concept of inadequate surveillance needs to be further explained in the introduction - why is this important for your approach to the analysis (threshold of API)? (lines 122-123)

Methods

- not sure what viz. (line 131) means and seen a few different times in the manuscript, please replace

- this is where is starts getting confusing for me. Why were these three levels of ABER chosen? How were these formulas developed and based on what information? As mentioned above, this relationship as formulated, needs to be substantiated based on preliminary descriptive work. And also, what is the end objective? If it's to understand how correlaed they are (or aren't) and how they diverge (at what thresholds of specific indicators), the current approach doesn't seem to address this

Results

- I would reduce the details in the results and focus on the main findings and make the graphs/tables central to the results section. It's too detailed oriented at the moment.

Discussion

- there should be more of a focus of how these results compare to other studies and contextualizing the reasons behind the study's findings

6. PLOS authors have the option to publish the peer review history of their article (what does this mean?). If published, this will include your full peer review and any attached files.

**Do you want your identity to be public for this peer review?** For information about this choice, including consent withdrawal, please see our Privacy Policy.

Reviewer #1: No

Reviewer #2: No

---

## [Decision Letter · Decision Letter 1]

17 Oct 2022

PGPH-D-22-00177R1

India may need an additional metric to assess the endemicity of malaria in low surveillance districts

Dear Dr. Sharma,

Thank you for submitting your manuscript to PLOS Global Public Health. After careful consideration, we feel that it has merit but does not fully meet PLOS Global Public Health’s publication criteria as it currently stands. Therefore, we invite you to submit a revised version of the manuscript that addresses the points raised during the review process.

We look forward to receiving your revised manuscript.

Kind regards,

Louisa Alexandra Messenger, MSc, PhD

Academic Editor

Journal Requirements:

Additional Editor Comments (if provided):

Reviewers' comments:

Reviewer's Responses to Questions

**Comments to the Author**

1. If the authors have adequately addressed your comments raised in a previous round of review and you feel that this manuscript is now acceptable for publication, you may indicate that here to bypass the “Comments to the Author” section, enter your conflict of interest statement in the “Confidential to Editor” section, and submit your "Accept" recommendation.

Reviewer #2: (No Response)

2. Does this manuscript meet PLOS Global Public Health’s publication criteria? Is the manuscript technically sound, and do the data support the conclusions? The manuscript must describe methodologically and ethically rigorous research with conclusions that are appropriately drawn based on the data presented.

Reviewer #2: Yes

3. Has the statistical analysis been performed appropriately and rigorously?

Reviewer #2: (No Response)

4. Have the authors made all data underlying the findings in their manuscript fully available (please refer to the Data Availability Statement at the start of the manuscript PDF file)?

Reviewer #2: Yes

5. Is the manuscript presented in an intelligible fashion and written in standard English?

Reviewer #2: Yes

6. Review Comments to the Author

Reviewer #2: The authors have done well in addressing my previous comments, although I have a few minor remaining comments.

Introduction

- This could use some more work before it is suitable for publication.

- For example, the first paragraph is too long, suggest dividing in two and the combining the very small third paragraph in the introduction (perhaps after is too detailed in terms of the

- Some of the information in paragraph 4 (lines 89-96) could be included in a context section in the methods (first section/paragraph in the methods) where the context of the study is described, including the currently surveillance situation for malaria

- combine paragraph 3 (too short and not connected to another paragraph) to paragraph 6 (lines 101-106)

- paragraph from lines 117-127 - also containing too much detail for an introductory paragraph, some of this information could be put into the context paragraph in the methods

- I would reword the study aim instead to be (using a PICOT format) - to compare and contrast malaria endemicity indicators API, ABER, and SPR, in india.

Methods

- API equation, please include reference

- unclear why these thresholds were chosen for margins of error (5%, 15%)

Results

- reasonably good (line 211) - what does this mean? moderate?

- and the conjunction of SPR (line 241) - reword to be more clear

- In a nutshell (line 242) - not appropriate scientific language In summary? Overall?

Discussion

- Facet (line 313) - Not clear what you mean here, would suggest instead remove

7. PLOS authors have the option to publish the peer review history of their article (what does this mean?). If published, this will include your full peer review and any attached files.

**Do you want your identity to be public for this peer review?** For information about this choice, including consent withdrawal, please see our Privacy Policy.

Reviewer #2: No

---

## [Editor Report · Decision Letter 2]

20 Oct 2022

India may need an additional metric to assess the endemicity of malaria in low surveillance districts

PGPH-D-22-00177R2

Dear Dr. Sharma,

We are pleased to inform you that your manuscript 'India may need an additional metric to assess the endemicity of malaria in low surveillance districts' has been provisionally accepted for publication in PLOS Global Public Health.

Best regards,

Louisa Alexandra Messenger, MSc, PhD

Academic Editor
